# Green Synthesis of 2-Oxazolidinones by an Efficient and Recyclable CuBr/Ionic Liquid System via CO₂, Propargylic Alcohols, and 2-Aminoethanols



**Chao Bu** [1,2], **Yanyan Gong** [3], **Minchen Du** [1,2], **Cheng Chen** [1], **Somboon Chaemchuen** [1], **Jia Hu** [1,2], **Yongxing Zhang** [1,2], **Heriberto Díaz Velázquez** [4], **Ye Yuan** [1,*] **and Francis Verpoort** [1,2,5,6,*]

[1] State Key Laboratory of Advanced Technology for Materials Synthesis and Processing, Wuhan University of Technology, Wuhan 430070, China; buchao0507@gmail.com (C.B.); yuqingwen321@gmail.com (M.D.); chengchen@whut.edu.cn (C.C.); sama_che@hotmail.com (S.C.); liuxiao17612704929@gmail.com (J.H.); zyx18327006800@whut.edu.cn (Y.Z.)

[2] School of Materials Science and Engineering, Wuhan University of Technology, Wuhan 430070, China

[3] State Key Laboratory of Biobased Material and Green Papermaking, Qilu University of Technology, Shandong Academy of Sciences, Jinan 250353, China; kxf@whut.edu.cn

[4] Dirección de Investigación en Transformación de Hidrocarburos, Instituto Mexicano del Petróleo, Mexico City 07730, Mexico; hdiaz@imp.mx

[5] National Research Tomsk Polytechnic University, Lenin Avenue 30, 634050 Tomsk, Russia

[6] Global Campus Songdo, Ghent University, 119 Songdomunhwa-Ro, Yeonsu-Gu, Incheon 21985, Korea

[*] Correspondence: fyyuanye@whut.edu.cn (Y.Y.); francis.verpoort@ghent.ac.kr (F.V.); Tel.: +86-186-3580-0380 (Y.Y.); +86-150-7117-2245 (F.V.)

**Abstract:** With the aim of profitable conversion of carbon dioxide ($CO_2$) in an efficient, economical, and sustainable manner, we developed a CuBr/ionic liquid (1-butyl-3-methylimidazolium acetate) catalytic system that could efficiently catalyze the three-component reactions of propargylic alcohols, 2-aminoethanols, and $CO_2$ to produce 2-oxazolidinones and α-hydroxy ketones. Remarkably, this catalytic system employed lower metal loading (0.0125–0.5 mol%) but exhibited the highest turnover number (2960) ever reported, demonstrating its excellent activity and sustainability. Moreover, our catalytic system could efficiently work under 1 atm of $CO_2$ pressure and recycle among the metal-catalyzed systems.

**Keywords:** carbon dioxide chemistry; copper catalysis; synthetic methods; multicomponent reaction; cyclization

## 1. Introduction

Carbon dioxide ($CO_2$), as a potential inducement for the greenhouse effect, has caught great attention from governments and scientific institutions [1,2]. On the other hand, $CO_2$ behaves as a nontoxic, abundant, easily accessible, and renewable C1 source, which is considered as an ideal feedstock for the construction of fine chemicals [3–9], fuels [10–13], polymers [14–16], etc. Hence, the strategy of $CO_2$ capture and utilization (CCU) came up, which aimed at the profitable conversion rather than the unhelpful storage after $CO_2$ was captured [17–24]. In this area, a rational choice for the absorbents that are used to fix $CO_2$ as well as induce the following conversion is vitally important [25–31]. Particularly, amino alcohols are considered as one of the most effective options due to the advantages of economy, low toxicity, strong absorption of $CO_2$, excellent stability of corresponding products, etc. [32–39]. Therefore, CCU strategies designed based on the various amino alcohols/$CO_2$ systems are highly promising. Particularly, the condensation of 2-aminoethanols with $CO_2$ attracted our attention because the corresponding product, 2-oxazolidinone, is one of the most important heterocyclic compounds that can be widely used as chemical intermediates [33,35,40–44], antibacterial drugs [45–47], etc. Unfortunately, this condensation

is generally incomplete due to the chemical equilibrium between the substrates of 2-aminoethanols and $CO_2$ and the products of 2-oxazolidinones and $H_2O$, which largely limits its practical application [35]. In order to solve this problem, dehydrating agents such as the traditional strong organic bases or electrophiles could be employed to shift the equilibrium toward the products [48–51]. However, this method inevitably consumed extra additives and generated unfavorable byproducts during the process. Other reports to overcome the thermodynamic barrier were also reported, such as the application of $CeO_2$ [52] or chlorostannoxane catalysts [53]. However, both of these processes required quite harsh reaction conditions (>150 °C) and the yields of 2-oxazolidinones were generally unsatisfactory.

Besides the efforts on direct condensation, researchers also developed alternative strategies that tried to circumvent the thermodynamic barrier of generating $H_2O$. Among them, employing propargylic alcohols in the condensation of 2-aminoethanols and $CO_2$ is a promising way that has been revealed as a thermodynamically feasible process. Moreover, $\alpha$-hydroxyl ketones, a series of high-value compounds that are generally employed as key synthons for organic chemistry and biologically active fragments in pharmacological products, could be simultaneously synthesized together with 2-oxazolidinones in this three-component process [54–58]. In this area, He et al. have achieved several milestones. Firstly, they employed 5 mol% of $Ag_2CO_3$ and 10 mol% of phosphine ligands (Xantphos) for this reaction, which could efficiently catalyze diverse substrates in $CHCl_3$ at 60 °C under 1 MPa of $CO_2$ [54]. Subsequently, a similar system containing 5 mol% of $Ag_2O$ and 30 mol% of 1,1,3,3-tetramethylguanidine was reported, which performed excellent activity under 1.0 MPa of $CO_2$ at 80 °C in $CH_3CN$ [55].

In addition to the silver catalytic systems, they also established a cheaper and greener Cu(I) catalytic system, in which a competitive amount of CuI (5 mol%) was added together with 5 mol% of 1,10-phen and 10 mol% of *t*-BuOK [56]. This system could promote the three-component reaction under a relatively low $CO_2$ pressure (0.5 MPa) at 80 °C. Recently, they synthesized a task-specific ionic liquid (IL), namely 1,5,7-triazabicyclo[4.4.0]dec-5-ene trifluoroethanol ([TBD][TFE]), which could work under 1 atm of $CO_2$ pressure at 80 °C [57]. Although great progress has been achieved for this strategy, several problems remained that blocked its further applications. For example, the only report of the metal-free catalyst [TBD][TFE] gave an acceptable catalytic performance, however, it was not commercially available and could be only obtained in laboratories by employing a rare organic base (TBD) through an anion exchange resin, which limited its large-scale application. In contrast, the metal-catalyzed systems employed simple and easily accessible materials as the catalysts, thus showing certain potential for practical applications. However, they still suffered from the disadvantages of high metal loading; elevated $CO_2$ pressure; poor catalyst recyclability; and additions of ligands, bases, or other additives. Consequently, developments of simple, green, easily accessible, and recyclable catalytic systems that perform excellent activity under mild conditions are still highly desirable.

Generally, IL is considered an environmentally friendly and green solvent for its negligible vapor pressure as well as high thermal stability. Particularly, its physical and chemical properties can be easily adjusted by changing the cations and anions or introducing desired functional groups, which largely extend its availability in diverse fields such as gas adsorption, catalysis, extraction, sample preparation techniques, etc. Therefore, employment of IL together with the metal salts might be a potential methodology to develop the desired catalytic systems. Herein, we combined the green and versatile Cu salts with the commercially available imidazole-based ILs for the three-component reactions of propargylic alcohols, 2-aminoethanols, and $CO_2$. After screening, an optimal CuBr/1-butyl-3-methylimidazolium acetate ([$C_4C_1$im][OAc]) catalytic system was obtained. This system proved to inherit the merits from both ILs and metal-catalyzed systems, which could efficiently promote the reaction under 1 atm of $CO_2$ pressure with a lowermost metal loading in the absence of any ligands, bases, and additives. Moreover, this system behaved

robustly in recyclability and sustainability. An unprecedented turnover number (TON) was achieved in this aspect.

## 2. Results and Discussion

Table 1 describes the screening of catalytic systems for the three-component reaction, including copper salts and ionic liquids. 2-(benzylamino)ethanol (1a) and 2-methylbut-3-yn-2-ol (2a) were used as the model substrates in the screening of the optimal catalytic systems for the three-component reaction, 3-(phenylmethyl)-2-oxazolidinone (3a) and 3-hydroxy-3-methyl-2-butanone (4a) as products of 1a and 2a (Table 1). First, blank experiments were performed, which showed that this reaction would not happen without catalysts (entries 1–3). However, considerable yields of 3a and 4a would be smoothly obtained under the catalysis of the CuBr/[$C_2C_1$im][OAc] system (entry 4). Subsequently, diverse Cu salts, such as CuCl, CuI, $Cu_2S$, Cu($CH_3CN$)$_4PF_6$, $C_4H_3S$-COO-Cu (Copper(I) thiophene-2-carboxylate), CuSCN, and CuOAc were employed together with the IL of [$C_2C_1$im][OAc] (entries 5–11). The experimental results showed that all these Cu salts exhibited considerable activity for this three-component reaction. Among them, CuBr gave the highest yield (entry 4). Afterward, different ILs varied in cations and anions were examined for their catalytic activity together with the optimal CuBr salt (entries 12–21). In the study of the anions, it could be clearly observed that $ClO_4^-$, $I^-$, $BF_4^-$, $PF_6^-$, and $OTf^-$ could not promote the model reaction (entries 12–16). While $Br^-$ and $NO_3^-$ gave detectable but much lower yields than $OAc^-$ (entries 17 and 18 vs. entry 4). On the other hand, [$C_4C_1$im]$^+$ and [$C_2C_1$im]$^+$, which discriminatively represented butyl- and ethyl-substituted imidazole-derived cations gave similar catalytic performances (entry 19 vs. entry 4). While for the combinations of $OAc^-$ with other kinds of cations such as [$N_{4444}$]$^+$ and [DBUH]$^+$, lower yields were obtained than for the imidazole-derived ones (entries 20 and 21 vs. entries 4 and 19). In general, butyl-substituted ILs were more economical and widely used than the ethyl ones. Thus, [$C_4C_1$im][OAc] was finally selected as the best IL. In summary, the combination of CuBr and [$C_4C_1$im][OAc] was considered to be the optimal catalytic system for the model three-component reaction (entry 19).

**Table 1.** Screen of the catalytic systems [a].

| Entry | [Cu] Salt | Ionic Liquid | Yield (%) [b] | |
|---|---|---|---|---|
| | | | 3a [b] | 4a [b] |
| 1 | - | [$C_2C_1$im][OAc] | 0 | 0 |
| 2 | CuBr | - | 0 | 0 |
| 3 | - | - | 0 | 0 |
| 4 | CuBr | [$C_2C_1$im][OAc] | 59 | 55 |
| 5 | CuCl | [$C_2C_1$im][OAc] | 56 | 50 |
| 6 | CuI | [$C_2C_1$im][OAc] | 55 | 56 |

**Table 1.** *Cont.*

| Entry | [Cu] Salt | Ionic Liquid | Yield (%) [b] | |
|---|---|---|---|---|
| | | | 3a [b] | 4a [b] |
| 7 | $Cu_2S$ | $[C_2C_1im][OAc]$ | 22 | 18 |
| 8 | $Cu(CH_3CN)_4PF_6$ | $[C_2C_1im][OAc]$ | 28 | 30 |
| 9 | $C_4H_3S\text{-}COO\text{-}Cu$ | $[C_2C_1im][OAc]$ | 55 | 57 |
| 10 | CuSCN | $[C_2C_1im][OAc]$ | 36 | 30 |
| 11 | CuOAc | $[C_2C_1im][OAc]$ | 17 | 14 |
| 12 | CuBr | $[C_2C_1im][ClO_4]$ | 0 | 0 |
| 13 | CuBr | $[C_2C_1im]I$ | 0 | 0 |
| 14 | CuBr | $[C_2C_1im][BF_4]$ | 0 | 0 |
| 15 | CuBr | $[C_2C_1im][PF_6]$ | 0 | 0 |
| 16 | CuBr | $[C_2C_1im][OTf]$ | 0 | 0 |
| 17 | CuBr | $[C_2C_1im]Br$ | 20 | 24 |
| 18 | CuBr | $[C_2C_1im][NO_3]$ | 24 | 27 |
| 19 | CuBr | $[C_4C_1im][OAc]$ | 60 | 60 |
| 20 | CuBr | $[N_{4444}][OAc]$ | 48 | 39 |
| 21 | CuBr | $[DBUH][OAc]$ | 37 | 35 |

[a] Unless otherwise specified, all the reaction conditions were as follows: **1a** (756.1 mg, 5 mmol, 1 equiv.), **2a** (630.9 mg, 1.5 equiv.), [Cu] (0.025 mmol, 0.5 mol%), ionic liquids (ILs) (6.5 mmol), at 80 °C under 0.1 MPa of $CO_2$ for 12 h. [b] Determined by $^1H$ NMR with 1,3,5-trimethoxybenzene as the internal standard.

After obtaining the best $CuBr/[C_4C_1im][OAc]$ system, we continued to optimize its condition parameters (Table 2). The reaction temperature was initially evaluated. In the beginning at 25 or 50 °C, the system was inactive without any products obtained (entries 1 and 2). However, the catalytic activity would increase along with the rising temperature from 50 to 100 °C (entries 2–4). A higher temperature of 120 °C was also tested; however, no obvious gain on the activity was observed (entry 5). Therefore, the suitable temperature was selected as 100 °C (entry 4). Furthermore, different amounts of $[C_4C_1im][OAc]$ and CuBr were also tried. Surprisingly, increasing or decreasing the IL would lead to reduced yields (entries 6–7 vs. entry 3). Meanwhile, a lower CuBr loading of 0.25 mol% showed an unsatisfactory yield (entry 8). Due to 0.5 mol% of CuBr had given a satisfactory result under 1 bar of $CO_2$, higher metal loadings or elevated $CO_2$ pressure were not further investigated. Lastly, the ratio of 1a:2a was tuned to 1:1 while the yield was decreased (entry 9), indicating an excess amount of propargylic alcohols would be beneficial for this reaction. In conclusion, the most suitable reaction conditions were fixed as follows: 0.5 mol% of CuBr and 1.3 equiv. of $[C_4C_1im][OAc]$ (based on 2-aminoethanols) under atmosphere $CO_2$ pressure at 100 °C with the ratio of 1:1.5 (1a:2a) (entry 4). It is worth noting that 0.5 mol% is the lowest metal loading ever reported among the metal-catalyzed systems, even the generally more active Ag catalysts could not reach this level. Meanwhile, this is the first reported metal-catalyzed system that could efficiently work under 1 atm of $CO_2$ pressure. Additionally, an experiment under the optimal conditions but without purging the system was performed; however, only moderate yields could be obtained (entry 10), indicating that lower $CO_2$ partial pressure or lower $CO_2$ purity was unfavorable for the reaction. Meanwhile, the purge operation was indeed necessary for obtaining high yields.

**Table 2.** Screen reaction conditions [a].

| Entry | CuBr (mol%) | [$C_4C_1$im][OAc] (mmol) | Temperature (°C) | The Ratio of 1a:2a | Yield (%) [b] | |
|---|---|---|---|---|---|---|
| | | | | | 3a [b] | 4a [b] |
| 1 | 0.5 | 6.5 | 25 | 1:1.5 | 0 | 0 |
| 2 | 0.5 | 6.5 | 50 | 1:1.5 | 0 | 0 |
| 3 | 0.5 | 6.5 | 80 | 1:1.5 | 60 | 60 |
| 4 | 0.5 | 6.5 | 100 | 1:1.5 | 92 | 96 |
| 5 | 0.5 | 6.5 | 120 | 1:1.5 | 90 | 95 |
| 6 | 0.5 | 3.2 | 80 | 1:1.5 | 27 | 27 |
| 7 | 0.5 | 13 | 80 | 1:1.5 | 55 | 58 |
| 8 | 0.25 | 6.5 | 80 | 1:1.5 | 20 | 23 |
| 9 | 0.5 | 6.5 | 80 | 1:1 | 44 | 48 |
| 10 [c] | 0.5 | 6.5 | 100 | 1:1.5 | 40 | 43 |

[a] Unless otherwise specified, all the reaction conditions were as follows: **1a** (756.1 mg, 5 mmol), under 0.1 MPa of $CO_2$ for 12 h. [b] Determined by $^1$H NMR with 1,3,5-trimethoxybenzene as the internal standard. [c] Without purging the system.

After obtaining the suitable catalytic system as well as its optimal reaction conditions, we started to explore the substrate scope. The experimental data are listed in Table 3. Initially, different propargylic alcohols substituted by the alkyl, cycloalkyl, and aryl groups (2a–2e) were examined. Delightfully, all these substrates could be transformed into the desired products at satisfactory yields. Specifically, 2d or 2e with relatively bulky substituent groups required prolonged time for the conversion, implying that the steric effects of the substituents might influence the reactivity of the propargylic alcohols. On the other hand, a series of 2-aminoethanols were also introduced into the reaction (1a–1j). Obviously, the substituents in the phenyl rings would also affect the reactivity of those substrates containing aryl groups. Generally, aryl 2-aminoethanols with electron-donating groups such as -Me or -MeO would smoothly accomplish the reaction, while the electron-withdrawing group $NO_2^-$ in 1f largely limits its reactivity for this reaction (1a–1d vs. 1f). In addition, alkyl substituted 2-aminoethanols, 1g–1j were also applied to the reaction, and moderate to excellent yields could be obtained, indicating the broad substrate scope of this catalytic system. Furthermore, a gram-scale experiment was performed based on 1a and 2a. The result showed that the CuBr/[$C_4C_1$im][OAc] system still exhibited satisfactory activity for grams of substrates, implying its potential in practical applications.

Besides catalytic activity, recyclability and sustainability were also important for comprehensively evaluating a catalyst. Herein, we explored the performance of the CuBr/[$C_4C_1$im][OAc] system in this aspect based on the model reaction of 1a and 2a under its optimal conditions. Owing to the advantage of the IL component that would retain the Cu salt during the extraction and separation, this catalytic system kept its excellent activity in the recycling assessment (as shown in Figure 1a), reflecting its stability and reusability (Table S2, supporting information). It is worth mentioning that this is the first metal-catalyzed system that could be reused for this three-component reaction. Subsequently, an experiment for evaluating the maximum turnover number (TON) was performed. To our delight, even when the metal loading reduced to an unprecedented level of 125 ppm, this catalytic system still exhibited considerable activity. Particularly, a TON of 2960 was obtained in this experiment (Figure 1b), indicating the excellent sustainability of this catalytic system. To our best knowledge, this is the highest TON ever reported for this three-component reaction (Figure S1 and Table S1, supporting information).

**Table 3.** Screening of the substrates [a].

| Entry | Substrate | | Product (Yield/%) [b] | | | |
|---|---|---|---|---|---|---|
| | 1 | 2 | 3 | | 4 | |
| 1 | 1a | 2a | 3a | 92 90 [c] 83 [d] | 4a | 96 93 [c] 78 [d] |
| 2 | 1a | 2b | 3a | 84 | 4b | 90 |
| 3 | 1a | 2c | 3a | 86 80 [d] | 4c | 85 79 [d] |
| 4 | 1a | 2d | 3a | 94 [e] | 4d | 98 [e] |
| 5 | 1a | 2e | 3a | 84 [f] 79 [d] | 4e | 85 [f] 80 [d] |
| 6 | 1b | 2a | 3b | 87 80 [d] | 4a | 91 77 [d] |
| 7 | 1c | 2a | 3c | 83 [e] | 4a | 89 [e] |
| 8 | 1d | 2a | 3d | 90 [e] | 4a | 89 [e] |

**Table 3.** *Cont.*

| Entry | Substrate | | Product (Yield/%) [b] | |
|---|---|---|---|---|
| | **1** | **2** | **3** | **4** |
| 9 | **1e** | **2a** | **3e** 85 | **4a** 90 |
| 10 | **1f** | **2a** | **3f** 17 [e] | **4a** 23 [e] |
| 11 | **1g** | **2a** | **3g** 51 [g] | **4a** 50 [g] |
| 12 | **1h** | **2a** | **3h** 77 [e] | **4a** 80 [e] |
| 13 | **1i** | **2a** | **3i** 73 [e] | **4a** 76 [e] |
| 14 | **1j** | **2a** | **3j** 92 [e] | **4a** 93 [e] |

[a] Unless otherwise specified, all the reaction conditions were as follows: CuBr (0.5 mol%), [C$_4$C$_1$im][OAc] (1.3 equiv.), **1** (5 mmol), **2** (1.5 equiv.), at 100 °C under 0.1 MPa of CO$_2$, 12 h. [b] Determined by $^1$H NMR with 1,3,5-trimethoxybenzene as the internal standard. [c] Gram-scale experiment: **1a** (1.5122 g, 10 mmol), **2a** (1.2618 g, 1.5 equiv.), [Cu] (0.05 mmol, 0.5 mol%), [C$_4$C$_1$im][OAc] (13 mmol), at 100 °C under 0.1 MPa of CO$_2$, 12 h. [d] Isolated yield. [e] 24 h. [f] 36 h. [g] CuBr (1 mol%).

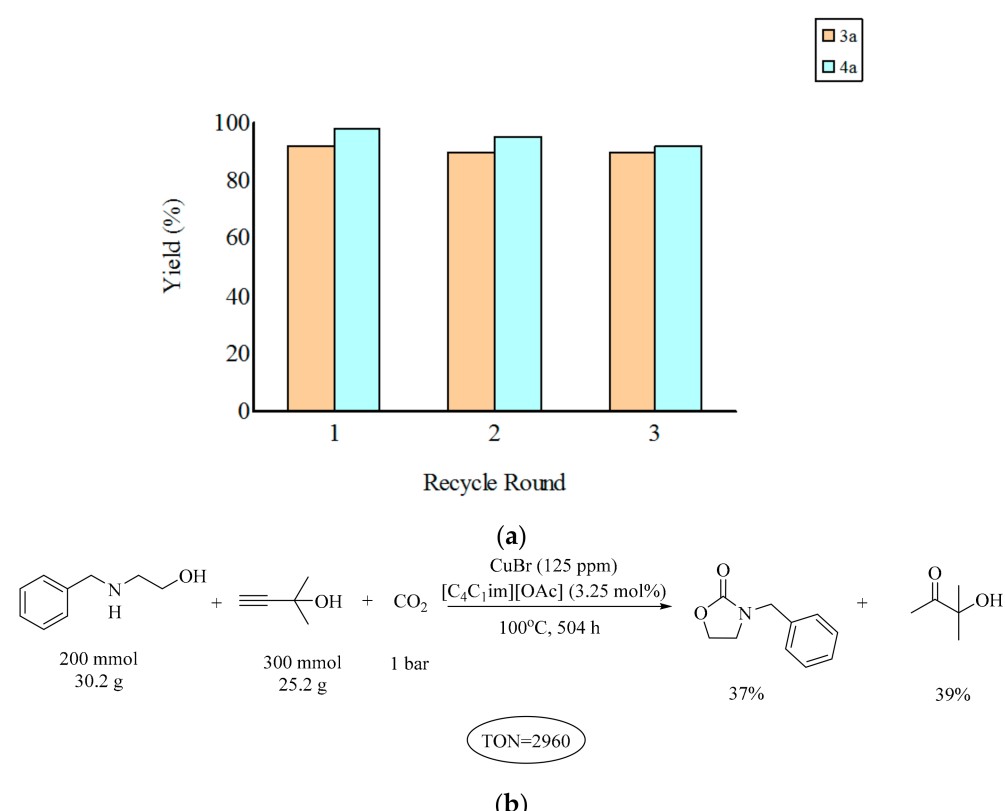

**Figure 1.** (**a**) Recyclability of the CuBr/[C₄C₁im][OAc] system; (**b**) evaluation of turnover number (TON) for the CuBr/[C₄C₁im][OAc] system.

## 3. Investigation of the Mechanism

### 3.1. Activation of the Hydroxyl Group

According to the previous literature, activation of hydroxyl groups in propargylic alcohols is the initial step of the three-component reaction, which could be monitored by the shape and chemical shift of the hydroxyl signal in $^1$H NMR [55,59,60]. Generally, this weak acidic proton of the hydroxyl group required relatively strong bases to activate it [61,62], and the OAc⁻ in normal acetate salts could not afford this activation [63]. However, from the following experiment, we verified that OAc⁻ in [C₄C₁im][OAc] could effectively activate the hydroxyl group.

Firstly, substrate 2a, and the mixture of 2a/[C₄C₁im][OAc] (1.5:1.3), 2a/1a (1.5:1) were respectively prepared in the closed Schlenk tubes at 100 °C. After 5 min, three samples were respectively taken from them into DMSO-$d_6$ and examined by $^1$H NMR (Figure 2). In Figure 2a, a sharp peak appeared at δ = 5.27 ppm, which was considered as the unactivated hydroxyl proton of the hydroxyl group. In the mixture of 2a/[C₄C₁im][OAc], the peak around 5.27 ppm became broad and shifted, confirming that the hydroxyl group was effectively activated with the aid of [C₄C₁im][OAc] (Figure 2b). However, in the 2a/1a system, the sharp peak was still maintained, indicating that 2-aminoethanol was invalid for this activation (Figure 2c). Interestingly, once $CO_2$ was introduced into the 2a/1a system, the hydroxyl peak was changed into a relatively obtuse shape, implying 2-aminoethanol together with $CO_2$ also showed slight activated ability for the hydroxyl proton (Figure 2d and Figure S4 of the supporting information). In consequence, [C₄C₁im][OAc] plays a vital role in the activation of the hydroxyl group, which initiates the following proposed mechanism.

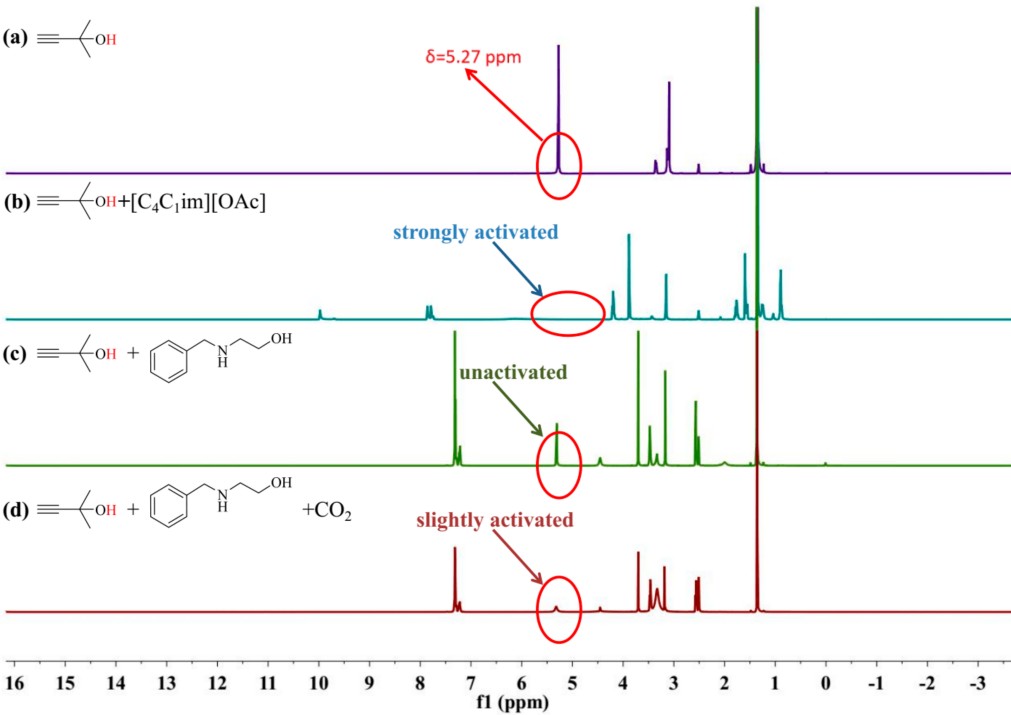

**Figure 2.** Investigations on the activation of hydroxyl protons.

### 3.2. Proposed Catalytic Mechanism

According to the previous publications [25,54,55,57,63–67], a probable catalytic mechanism of the CuBr/[$C_4C_1$im][OAc] system was proposed for the three-component reaction (Scheme 1a), which might contain two steps: (1) propargylic alcohols are combined with $CO_2$ to generate the key cyclic carbonates, D; (2) D react with aminoethanols to give 2-oxazolidinones and $\alpha$-hydroxyl ketones (Figures S2 and S3, supporting information). In step 1, the OAc$^-$ anion initially activates the hydroxyl group of the propargylic alcohol and $CO_2$ [68,69], which is favorable for the following attack of the hydroxyl oxygen to the carbon center of the $CO_2$, generating intermediate B. Then, the metal catalyst activates the triple bond so that the negative oxygen in intermediate B can attack the carbon of this triple bond intramolecularly and form intermediate C. Finally, the catalyst is released from the five-membered ring through the returning of the proton, giving the important intermediate cyclic carbonate D. Then step 2 occurs, in which the nitrogen of the aminoethanol attacks the carbon in D and breaks the C–O bond, resulting in the breakage of the five-membered ring and the generation of E. E is converted to F due to its unstable enol structure. Finally, the hydroxyl oxygen attacks the adjacent carbonyl carbon with the aid of the catalysts. A five-membered ring of 2-oxazolidinone is generated by releasing an $\alpha$-hydroxy ketone molecule.

Interestingly, besides the general mechanism of the Cu salt, another Cu species might also exist in our catalytic system. According to our previous reports [25,67], the basic OAc$^-$ in [$C_2C_1$im][OAc] might interact with the imidazole cation, leading to the chemical equilibrium with the free N-heterocyclic carbenes (NHCs) and the corresponding HOAc. Once Ag salts are involved, the NHCs might be coordinated in situ and form the NHC–Ag complexes. Therefore, we speculated that similar NHC–Cu complexes might also exist in this Cu-catalyzed system (Scheme 1b).

**Scheme 1.** (**a**) Proposed catalytic mechanism of the CuBr/[C$_4$C$_1$im][OAc]; (**b**) possible generation of the N-heterocyclic carbene (NHC)–Cu complexes.

### 3.3. Exploration of the NHC–Cu Complexes

Firstly, the following experiment was performed: 5 mmol of 1a and 7.5 mmol of 2a were catalyzed by 0.5 mmol CuBr/6.5 mmol [C$_4$C$_1$im][OAc] at 100 °C under 0.1 MPa of CO$_2$ for 3 h. Once this reaction finished, the obtained mixture was sampled and analyzed directly by High-Resolution Mass Spectrometry (HRMS) (Figure 3). From the spectrum, a signal of 339.16040 and another three signals of 340.16364, 341.15816, and 342.16267 were respectively observed, which matched with the exact mass and the corresponding isotopes of the bis-NHC–Cu complex (Scheme 1b). On the other hand, no signal of mono-NHC–Cu was detected in the HRMS spectrum. This result confirmed the existence of the NHC–Cu complexes in the catalytic reaction, which matched the bis-NHC–metal structure.

Subsequently, based on the experimental results and our previous study [67], we speculated a probable mechanism involving the bis-NHC–Cu complex (Scheme 2). The main parts were consistent with the mechanism in Scheme 1a. Particularly, when the bis-NHC–Cu complex enters the catalytic cycle, one NHC might drop and participate in the interaction between OAc$^-$ and the hydroxyl proton. Meanwhile, the remaining [Cu] species perform the same role as the normal Cu salt.

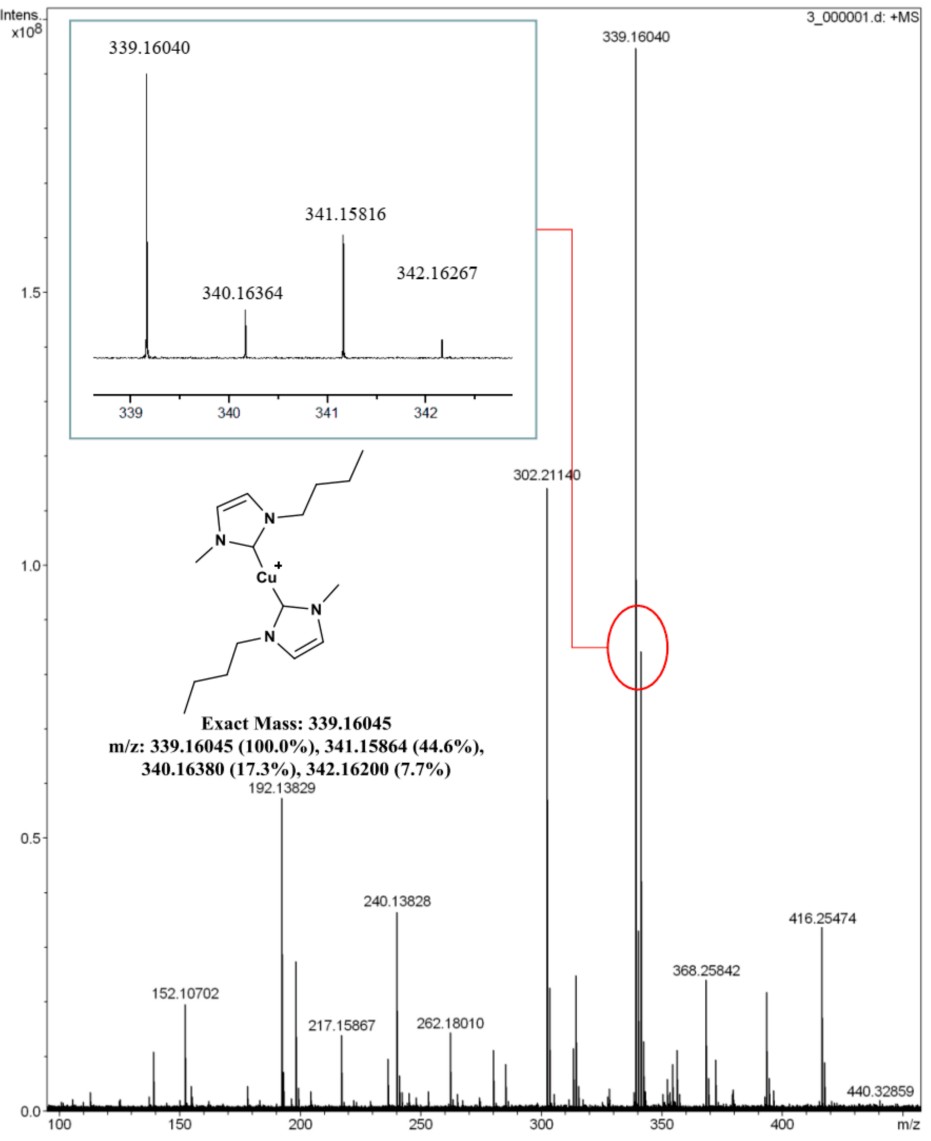

**Scheme 2.** Proposed catalytic mechanism involving the bis-NHC–Cu complex.

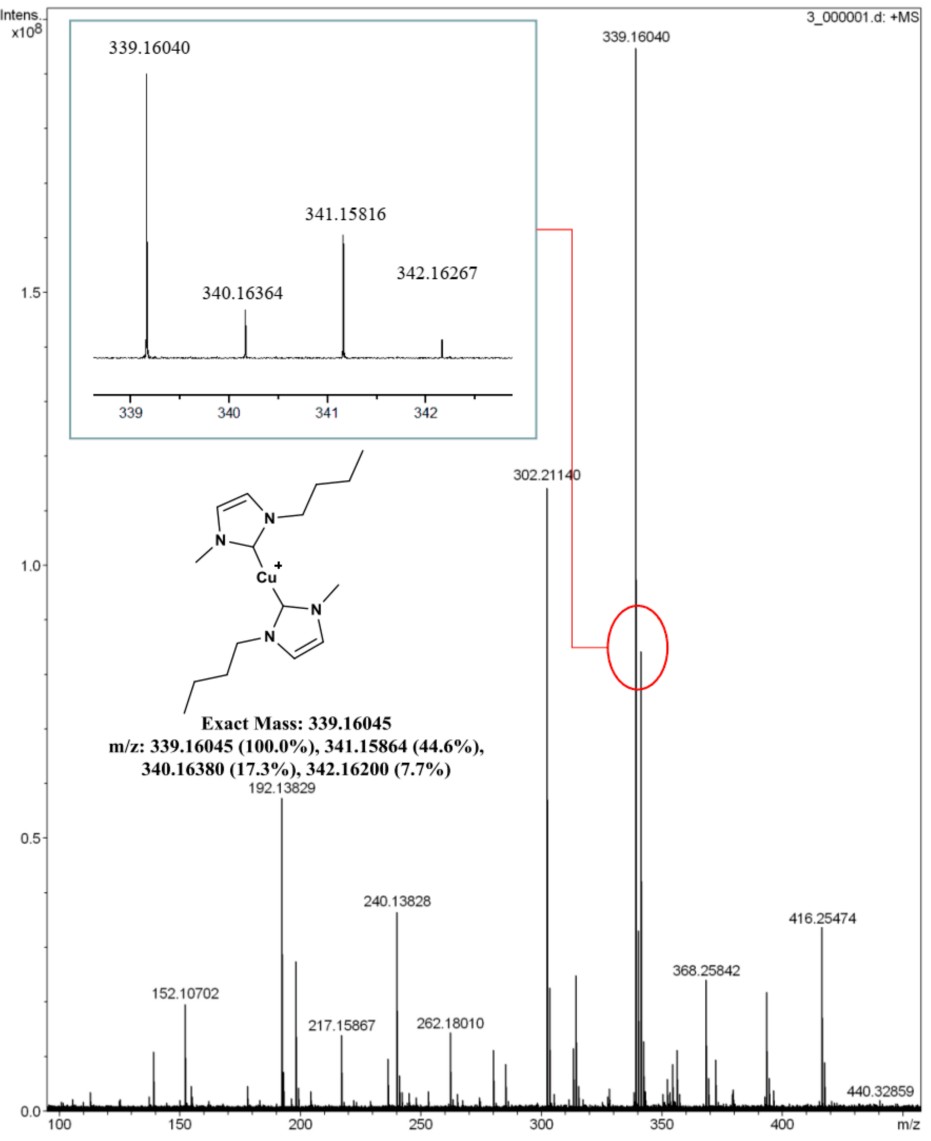

**Figure 3.** High-Resolution Mass Spectrum of the system of reaction mixtures.

## 4. Materials and Methods

### 4.1. Characterization

All the nuclear magnetic spectra were obtained by a Bruker Avance III HD spectrometer. $^1$H NMR was recorded at 500 MHz in CDCl$_3$ (7.26 ppm) or DMSO-$d_6$ (2.51 ppm), and $^{13}$C NMR was recorded at 126 MHz in CDCl$_3$ (77.16 ppm) or DMSO-$d_6$ (39.52 ppm). High-resolution mass spectra were conducted by a Bruker Daltonics micro TOF-QII mass spectrometry instrument given in per charge ($m/z$).

### 4.2. Materials

CO$_2$ at a purity of 99.999% was purchased from the Xiang Yun Gas Company. Unless specifically mentioned, all the raw materials, including propargylic alcohols, copper salts, and ionic liquids, were obtained from Sigma-Aldrich, Aladdin, TCI, Macklin, Alfa Aesar, etc. [DBUH][OAc] [59] and 2-aminoethanols [54,55] (except 1a, 1g–1j) were synthesized following the reported literatures.

### 4.3. Three-Component Reactions of Propargylic Alcohols, 2-Aminoethanols, and CO$_2$

Propargylic alcohols (7.5 mmol), 2-aminoethanols (5 mmol), CuBr (0.025 mmol), and [C$_4$C$_1$im][OAc] (6.5 mmol) were added into a reaction tube equipped with a magnet bar. The gas inside the tube was replaced by CO$_2$ (99.999%) three times to confirm that this system was completely under the atmosphere of 1 atm of CO$_2$. Then the tube was heated in an oil pot at 100 °C for 12 h. When the reaction was completed, the mixture was extracted by diethyl ether (5 × 10 mL). Finally, the upper layers were collected and evaporated by the rotary evaporator. The obtained raw products were further separated and purified by column chromatography. For the recyclability investigation, the lower layer (recovered CuBr and [C$_4$C$_1$im][OAc]) was directly reused for the next round after drying under vacuum at 100 °C for 3 h.

## 5. Conclusions

In summary, we have developed a CuBr/[C$_4$C$_1$im][OAc] catalytic system that can efficiently produce 2-oxazolidinones and α-hydroxy ketones through the three-component reactions of propargylic alcohols, 2-aminoethanols, and CO$_2$ in a convenient and green manner. Particularly, this system exhibited excellent catalytic activity under 1 bar of CO$_2$ with only 0.0125–0.5 mol% of CuBr. Furthermore, the robust recyclability and sustainability of this system were also demonstrated with an unprecedented TON of 2960, the highest ever reached. In further mechanistic investigations, we detected an NHC–Cu complex during the experimental process, which was eventually identified as a bis-NHC–Cu configuration by the HRMS.

**Supplementary Materials:** The following are available online at https://www.mdpi.com/2073-4344/11/2/233/s1, Figure S1. The literatures reported for the three-component reactions; Figure S2. 1H NMR of the control experiment mixture (red) and the pure cyclic carbonate (blue); Figure S3. 1H NMR of pure 4a (green), pure 3a (red) and the control reaction mixture (blue); Figure S4. Investigations on the activation of hydroxyl protons in the presence of 1 atm of CO$_2$; Table S1. TON reported in the previous literatures; Table S2. Exploration of metal leaching in the recycling experiments.

**Author Contributions:** C.B.: investigation, experiments, writing—original draft. Y.G.: investigation, experiments. M.D.: investigation, experiments. C.C.: writing—review and editing. S.C.: writing—review and editing. J.H.: investigation. Y.Z.: investigation. H.D.V.: writing—review and editing. Y.Y.: methodology, supervision, writing—review and editing. F.V.: supervision, funding acquisition, conceptualization, writing—review and editing. All authors have read and agreed to the published version of the manuscript.

**Funding:** We appreciate the National Natural Science Foundation of China (no. 21950410754) and the Fundamental Research Funds for the Central Universities (no. 205201028, 205201026).

**Institutional Review Board Statement:** Not applicable.

**Informed Consent Statement:** Not applicable.

**Data Availability Statement:** The data presented in this study are available on request from the corresponding author.

**Acknowledgments:** The authors would like to express their deep appreciation to the State Key Lab of Advanced Technology for Materials Synthesis and Processing for their financial support.

**Conflicts of Interest:** The authors declare that they have no known competing financial interests or personal relationships that could have appeared to influence the work reported in this paper.

**Sample Availability:** Samples of the compounds are available from the authors.

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
