# Peer review of "Green Synthesis of 2-Oxazolidinones by an Efficient and Recyclable CuBr/Ionic Liquid System via CO2, Propargylic Alcohols, and 2-Aminoethanols"

_catalysts, doi:10.3390/catal11020233_

Round 1
Reviewer 1 Report
This MS concerning the Green Synthesis of 2-Oxazolidinones by an Efficient and Recy- 2lable CuBr/Ionic Liquid System via CO2, Propargylic alcohols, 3
and 2-Aminoethanols results really useful to the actual literature. The paper is generally convincing, I would just suggest a minor revision in order to correct some little things:
In Section 2 Results and discussions there shouldn't be the sections: characterization, materials, etc.. Indeed these section should be at the end of the MS in the section: materials and methods
In Section 3 a reference should be added. Indeed, it seems quite muddler to have just "3a and 4a". For example the sentence could be started explaining what is present is table 1 and then the discussion about it should follow.
Author Response
Point 1: In Section 2 Results and discussions there shouldn't be the sections: characterization, materials, etc.. Indeed these section should be at the end of the MS in the section: materials and methods
Response 1: As the reviewer suggested, we move this section to the end of the MS in the section: materials and methods and marked with yellow color
Point 2: In Section 3 a reference should be added. Indeed, it seems quite muddler to have just "3a and 4a". For example the sentence could be started explaining what is present is table 1 and then the discussion about it should follow.
Response 2: As the reviewer suggested, the sentence started explaining what is present in table 1 is added and marked with yellow color.
“…Table 1 describes the screening of catalytic systems for the three-component reaction, including copper salts and ionic liquids…”
Moreover, the exact names of 3a and 4a were also added.

Reviewer 2 Report
I recommend to accept the article in the present form.
Author Response
Reviewer 2:
I recommend to accept the article in the present form.
Response to reviewer #2:
We appreciate the time and efforts of referee 2 to review our manuscript.
Reviewer 3 Report
This manuscript describes the green synthesis of biologically and synthetically valuable compounds (2-oxazolidinones and alpha-hydroxy ketones) from a three-component reaction involving carbon dioxide and catalyzed by a copper I / ionic liquid system.
Compared to the results previously described in the literature, the reported research presents several important achievements. The catalytic system consists of a non-toxic and non-expensive transition metal (copper) which can be used with very low catalyst loadings (0.5 mol%), presents a high TON, and can be recycled. The three-component reaction can be applied to a wide range of substrates (both the aminoethanols and the propargylic alcohols) enabling to efficiently use carbon dioxide and to prepare two chosen valuable organic molecules. The reaction works with a low pressure of carbon dioxide (1 atm).
This article is well-written and the results are well-presented. The research work is very complete and presents, in complement to the screenings of the reaction conditions and the substrates, an investigation of the reaction mechanism. All the conclusions and the hypotheses made by the authors are clearly supported by the results. The experimental part in the supporting information is also very detailed, enabling the reader to reproduce the results, and the synthesized molecules are sufficiently characterized (all the NMR spectra are provided).
In my opinion, the achievements of this research (compared to previously described results) would be of great interest to the readers of Catalysts. I recommend accepting the manuscript for publication after some very minor corrections:
- Part 2 and Part 3 share the same title. Part 2 should be "Material and Methods"
- in Fig1b, there is an empty bubble under the reaction scheme, is this normal? What does it mean? Could you explain or correct it?
- in my opinion, the sole drawback of the reported synthesis is the use of an excess of propargyl alcohols, as it is not good for the atom economy, and in order to capture one carbon atom from CO2, it is necessary to "waste" a non-negligible quantity of propargylic alcohol. Did the authors try to recover the alkylidene cyclic carbonate? Can it be valorized? If this product evolves during treatment/purification steps, are the by-products valuable?
- In the supporting information, Fig S1, I think that the hydroxy ketones (in lines 2, 3, 4 and this work) do not correspond to the starting propargylic alcohols.
Author Response
Response to Reviewer 3 Comments
Point 1: in Fig1b, there is an empty bubble under the reaction scheme, is this normal? What does it mean? Could you explain or correct it? 

Response 1: Sorry, there was a “TON=2960” in this empty bubble (as showed below). A mistake might happen that made it invisible when converted the document to PDF.
Point 2: in my opinion, the sole drawback of the reported synthesis is the use of an excess of propargyl alcohols, as it is not good for the atom economy, and in order to capture one carbon atom from CO2, it is necessary to "waste" a non-negligible quantity of propargylic alcohol. Did the authors try to recover the alkylidene cyclic carbonate? Can it be valorized? If this product evolves during treatment/purification steps, are the by-products valuable?
Response 2:
Yes, we indeed tried to recover the alkylidene cyclic carbonates as the reviewer mentioned. However, we noticed that after reaction the amount of alkylidene cyclic carbonates was quite small, which was not deserved to recover. This might be due to the following reasons. First, it might be because of the volatilization of the propargyl alcohols. Although 1.5 equiv. of propargyl alcohols were employed, not all of them could participate in the reaction. Secondly, due to the limitation of the catalytic activity, not all of the left propargyl alcohols could fully be converted into the alkylidene cyclic carbonates.
In principle, alkylidene cyclic carbonates could be valorised into valuable products. However, because the amount of alkylidene cyclic carbonate was quite small after the main reaction, no valorised processes or by-products were clearly detected.
Point 3: In the supporting information, Fig S1, I think that the hydroxy ketones (in lines 2, 3, 4 and this work) do not correspond to the starting propargylic alcohols.
Response 3: As the reviewer suggested, the hydroxy ketones (in lines 2, 3, 4 and this work) had been changed to be the same as the starting propargylic alcohols and marked with yellow color.